# Role of ambient pressure in self-heating torrefaction of dairy cattle manure

Takanori Itoh[1¤], Kazunori Iwabuchi[1]*, Naohiro Maemoku[2], Siyao Chen[2], Katsumori Taniguro[3]

1 Research Faculty of Agriculture, Hokkaido University, Sapporo, Hokkaido, Japan, 2 Graduate School of Agriculture, Hokkaido University, Sapporo, Hokkaido, Japan, 3 Tanigurogumi Corporation, Nasushiobara, Tochigi, Japan

¤ Current address: Tanigurogumi Corporation, Nasushiobara, Tochigi, Japan
* iwabuchi@bpe.agr.hokudai.ac.jp

## Abstract

This paper describes the role of ambient pressure in self-heating torrefaction of livestock manure. We explored the initiating temperatures required to cause self-heating of wet dairy cattle manure at different ambient pressures (0.1, 0.4, 0.7, and 1.0 MPa). Then, we conducted proximate, elemental, and calorific analyses of biochar torrefied at 210, 250, and 290°C. The results showed that self-heating was induced at 155°C or higher for 0.1 MPa and at 115°C or lower for 0.4 MPa or higher. The decrease of the initiating temperature at elevated pressure was due not only to more oxygen, but also to the retention of moisture that can promote chemical oxidation of manure. Biochar yields decreased with increasing torrefaction temperature and pressure, and the yield difference at 0.1 and 1.0 MPa was more substantial at lower temperatures: a 29.8, 16.4, and 9.4% difference at 210, 250, and 290°C, respectively. Proximate and elemental analyses showed that elevated pressure promotes devolatilization, deoxygenation, and coalification compared to atmospheric pressure; its impact, however, was less at higher temperatures as the torrefaction temperature became more dominant. Calorific analysis revealed that elevated pressure can increase the higher heating value (HHV) on a dry and ash-free basis at 210°C because of the increase in carbon content, but its impact is limited at 250 and 290°C. Meanwhile, the HHV on a dry basis exhibited the opposite trend due primarily to an enlargement of ash content. The present study revealed that ambient pressure considerably affects the initiating temperature of self-heating and the chemical properties of biochar at a low torrefaction temperature.

## Introduction

Due to the growth of intensive farming, the thermochemical treatment of livestock manure is gathering attention in the livestock sector. Although manure can be used as an organic fertilizer and soil amendment, its overapplication may cause some environmental issues as it can be a primary source of greenhouse gas emissions, water pollution, and acidification [1–4]. In this context, the conversion of manure into biochar is an attractive option. Since biochar improves

**Data Availability Statement:** All data for this work are available from the Dryad Digital Repository at https://doi.org/10.5061/dryad.4j0zpc87r.

**Funding:** This study was supported by Tanigurogumi Corporation, Japan, and by the

Japan Society for the Promotion of Science (JSPS) KAKENHI (No. JP262692130 to K.I. and No. JP17J00272 to T.I.), https://www.jsps.go.jp/english/. The commercial funder provided support in the form of salaries for authors [T.I. and K.T.] and designed the study but did not have any additional role in the data collection and analysis, decision to publish, or preparation of the manuscript. The specific roles of these authors are articulated in the 'author contributions' section.

**Competing interests:** This study received financial support from Tanigurogumi Corporation, Japan. T. I. has been an employee of Tanigurogumi Corporation, Japan, since April 2019. K.T. is president of Tanigurogumi Corporation, Japan. There is no further employment, consultancy, patents, products in development, or marketed products to declare. This does not alter our adherence to PLOS ONE policies on sharing data and materials.

the physicochemical properties of the soil and retains nutrients, it is believed that the application of biochar to farmland has positive effects on crop growth [5–7]. Moreover, owing to the upgraded fuel properties, it can be utilized as a feedstock for gasification or co-firing [8–10]. Thus, biochar production from manure not only lowers the risk of environmental impacts, but also has economic benefits.

From the perspective of energy balance, the high moisture of manure makes it more difficult to perform dry torrefaction, which requires pre-drying before the slow pyrolysis step at 200–300˚C [11,12]. For this reason, wet torrefaction (also referred to hydrothermal carbonization), employing hot-compressed water of 180–260˚C and not requiring pre-drying, is often selected for feedstock with high moisture content [11,13]. It has been reported that wet torrefaction at a temperature of 250˚C for 480 min increases the higher heating value (HHV) of poultry litter from 17.2 to 25.2 MJ/kg [14]. In addition, microwave-assisted hydrothermal carbonization of dairy manure provides hydrochar with better chemical and structural properties [15]. Although these previous studies show the potential application of wet torrefaction for manure management, some technical issues remain, especially for the processed liquid. The liquid remaining after wet torrefaction contains organic compounds including furfural and its derivatives, organic acids, and phenol and phenolic derivatives [16], and chemical or biological treatment would be needed before discharging it into the environment [17–19].

To overcome this practical difficulty, we recently reported a self-heating torrefaction system that uses the heat generated from the chemical oxidation of manure as a heat source for drying and torrefaction [20]. In this system, the self-heating of the feedstock from below 100 up to 300˚C enables the conversion of wet manure into dry biochar. As self-heating torrefaction extracts the required energy for converting to biochar from feedstock, energy consumption will be reduced significantly. However, further understanding of the system is necessary before its implementation. Self-heating torrefaction contains several unique parameters, such as pre-heating temperature, airflow rate, and ambient pressure, which must be set appropriately to induce stable self-heating. Among these parameters, ambient pressure plays a significant role in the system. For instance, we confirmed that an elevated-pressure of 1.0 MPa induces self-heating at below 100˚C, while atmospheric pressure (0.1 MPa) does not [20], indicating that ambient pressure is highly relevant to the initiating temperature of self-heating. In addition, a few studies have reported that torrefaction under elevated pressure improves fuel properties such as fuel ratio and energy density [21,22]. These previous studies imply that the ambient pressure in self-heating torrefaction is associated with the chemical properties of the resulting char as well as the torrefaction temperature. In the present study, we investigated the role of ambient pressure in the self-heating torrefaction process and the chemical properties of the resulting biochar. We explored the initiating temperatures that can cause the stable self-heating of manure and observed the temperature profiles of the process at different pressures. Then, we investigated the pressure effects by characterizing the chemical properties of the biochar.

## Materials and methods

### Self-heating torrefaction system

Dairy cattle manure sampled from an experimental farm (Field Science Center for Northern Biosphere, Hokkaido University, Japan, latitude 43˚ 04′ N, longitude 141˚ 20′ E) was used in this study. The torrefaction system used in the present study was in accordance with a previous report [20]. Briefly, a 1-L stainless-steel reactor containing 200 g of wet sample was placed in an STPH-102M oven (Espec, Osaka, Japan). The air supply was provided by an air cylinder and F-201CV Series mass flow controller (Bronkhorst Japan, Tokyo, Japan). Ambient pressure

inside the reactor was maintained at given pressure with a BP-3 series back-pressure regulator (Go Regulator, Spartanburg, SC, USA). The temperatures of the sample and oven were recorded by K-type thermocouples. Once the sample temperature exceeded the preset temperature due to self-heating, the oven temperature was controlled to follow the sample temperature to within 1.5˚C to improve thermal insulation.

### Exploration of the initiating temperatures at different pressures

We investigated the initiating temperatures that can induce the self-heating of manure at different absolute pressures of 0.1, 0.4, 0.7, and 1.0 MPa, and at a constant ventilation rate of 0.8 $L_n$ min$^{-1}$ kg-AFS$^{-1}$ (AFS: ash-free solid). Self-heating was defined as follows: the increased rate of the sample temperature was greater than 10˚C in 24 h after the sample reached the preset temperature. This criterion was defined based on preliminary experiments that showed stable self-heating. The self-heating experiments were repeated at least twice to ascertain the experimental accuracy.

### Effects of temperature and pressure on biochar characteristics

After revealing the relationship between pressure and the initiating temperatures, we examined the effects of temperature and pressure on the characteristics of biochar. A full factorial experimental design was used with the torrefaction temperature and ambient pressure as two factors. A total of 12 treatments were designed: the three torrefaction temperature levels (210, 250, and 290˚C) and four absolute pressure levels (0.1, 0.4, 0.7, and 1.0 MPa). Based on the results obtained in previous experiments, a preheating temperature of 160˚C and a ventilation rate of 0.8 $L_n$ min$^{-1}$ kg-AFS$^{-1}$ were used to induce the self-heating of manure for all treatments. Since the self-heating of manure continues throughout the process, nitrogen gas was supplied to stop the reaction after the sample reached the desired torrefaction temperatures. Biochar preparation at each treatment was performed twice, and all reported data relating to biochar characteristics are expressed as the average values ± standard deviations of two replicates.

### Sample analyses

Proximate, ultimate, and calorific analyses were conducted to determine the fuel properties of biochar samples. The moisture and ash content of samples was measured by drying the subsamples at 105˚C for 24 h and by burning the oven-dried subsamples at 600˚C for 3 h in an FUL220FA electric muffle furnace (Advantec, Tokyo, Japan). The VM content was determined by establishing the mass loss resulting from heating the oven-dried subsamples at 950 ± 20˚C for 7 min in an ICKV electric furnace for VM determination (Ishizuka Electronic, Tokyo, Japan), in accordance with ASTM E872-82 [23], and the FC content was determined by the difference (FC = 100 − VM − Ash). The elemental contents of carbon (C), hydrogen (H), and nitrogen (N) were determined by a CE-440 elemental analyzer (Exeter Analytical, North Chelmsford, MA, USA). The oxygen content was calculated by the difference (O = 100 − C − H − N). The HHV was measured using an OSK 200 bomb calorimeter (Ogawa Sampling, Saitama, Japan), and data are expressed on a dry basis (HHV[db]) and dry ash-free basis (HHV[daf]). All measurements were repeated at least twice to ensure reproducibility.

### Statistical analyses

A total of 24 assays, corresponding to 12 treatments and two replicates, were used to conduct a two-way analysis of variance (ANOVA) with the torrefaction temperature and ambient

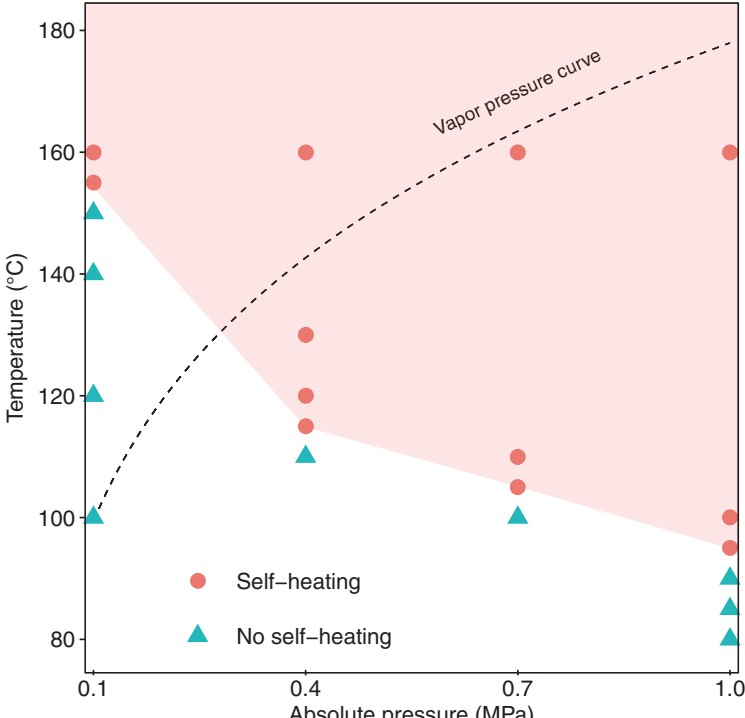

**Fig 1. Relationship between ambient pressure and initiating temperature of the self-heating of manure.** Self-heating is induced if the ambient pressure and preheating temperature in the red field are used.

pressure as factors. If there was an interaction between the two factors, *post hoc* Tukey's HSD test (α = 0.05) was employed to evaluate differences in mean values among treatments. Statistical analyses were carried out using R statistical software (version 3.6.2 for Mac OS X; https://www.r-project.org).

## Results and discussion

### Effects of ambient pressure on self-heating of manure

First, the influence of ambient pressure on the initiating temperature of the self-heating of manure was investigated (Fig 1). The initiating temperatures decreased with increasing pressure; atmospheric pressure (0.1 MPa) required a preheating temperature of 155˚C or higher to induce the self-heating of manure, while an elevated pressure of 1.0 MPa caused the process to proceed from 100˚C or lower. The drop of initiating temperatures can be explained by an increase of the dissolved oxygen amount in the water film covering the manure surface. Since the self-heating of manure is an oxidation process, the rate of the reaction is accelerated as the contact frequency between the manure and oxygen increases. We believe that an elevated-pressure environment has more ability to supply oxygen to the manure and thereby lower the initiating temperature. In addition to the increase in oxygen concentration, the moisture in the manure also plays an important role. Considering the saturated vapor curve, the elevated-pressure environment caused self-heating at below the boiling point of water (Fig 1), indicating that the self-heating of manure starts when the sample is wet. It has also been reported that the moisture in biomass can promote low-temperature oxidation of manure at 90˚C and chemical self-heating of solid waste [24,25]. Accordingly, an elevated-pressure environment helps retain

the moisture that promotes the oxidation of biomass, so that the self-heating of manure at lower temperatures is achieved.

Fig 1 provides the operating range where the self-heating of manure occurs steadily: ≥155˚C at 0.1 MPa, ≥115˚C at 0.4 MPa, ≥105˚C at 0.7 MPa, and ≥95˚C at 1.0 MPa. Based on these results, in the present study we used a preheating temperature of 160˚C to induce the self-heating of manure in the range of 0.1 to 1.0 MPa and investigated the effects of pressure on the chemical properties of the resulting chars. Fig 2 depicts the temperature profiles of the self-heating up to 290˚C at different pressures. Under all pressure conditions, we confirmed the rise of temperature up to 290˚C accompanied by temporary temperature stagnation. The stagnation occurred at around the boiling point of each pressure, indicating that the drying stage is dominant in that period. Considering the boiling point at each pressure, the preheating temperature of 160˚C was higher than the boiling point of 100 and 143˚C (at 0.1 and 0.4 MPa), while it was lower than those of 163 and 178˚C (at 0.7 and 1.0 MPa). This indicates that the drying process at 0.1 and 0.4 MPa was completed during the preheating stage, which is driven by external heating. In contrast, the drying process at 0.7 and 1.0 MPa was accomplished not by external heating but by self-heating of the manure, resulting in a longer period for drying than at 0.1 and 0.4 MPa. In the assumed system, we believe that external heating is not necessary once the self-heating initiates, so the amount of energy required for the entire system is estimated to be much smaller at 0.7 and 1.0 MPa than at 0.1 and 0.4 MPa. The above suggests that setting a preheating temperature below the boiling point at each pressure will improve the energy efficiency of the entire process, although such process optimization is outside of the scope of the present study.

## Biochar yield

Biochar yields at different torrefaction temperatures and ambient pressures are shown in Fig 3. Overall, the yields showed decreasing trends with increasing temperature and pressure, and showed a maximum of 91.4 ± 3.3%db for 210˚C at 0.1 MPa and a minimum of 53.3 ± 2.0%db for 290˚C at 1.0 MPa. It is well known that the torrefaction temperature considerably affects

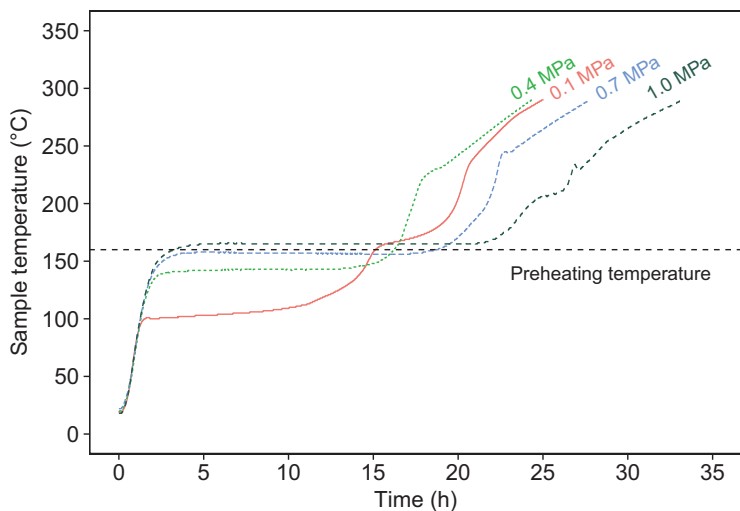

**Fig 2. Temperature profiles of self-heating of manure under different pressures.** A preheating temperature of 160˚C was used.

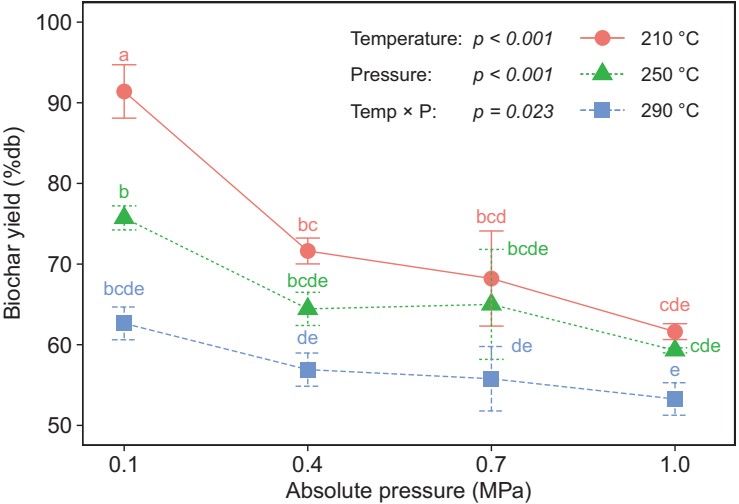

**Fig 3. Biochar yields under different torrefaction temperatures and ambient pressures.**

the biochar yield, and higher temperatures give lower yields. In addition, the results of the two-way ANOVA revealed that there is an interaction between the torrefaction temperature and ambient pressure that affects the biochar yield ($p = 0.023$) (Table 1). The influence of the pressure on the yield is substantial at lower temperatures and becomes less at higher temperatures. For instance, as the pressure increased from 0.1 to 1.0 MPa, the yield ranged from 91.4 to 61.6%db at 210˚C (29.8% difference, $p < 0.001$), from 75.7 to 59.3%db at 250˚C (16.4% difference, $p = 0.010$), and from 62.7 to 53.3%db for 290˚C (9.4% difference, $p = 0.271$).

Biomass consists of three main components, hemicellulose, cellulose, and lignin, and the thermal decomposition of these three components happens during torrefaction. The thermal

**Table 1. Two-way analysis of variance with torrefaction temperature and ambient pressure as two factors for 16 responsible variables.**

| Response variables | Temperature | Pressure | Temperature × Pressure |
|---|---|---|---|
| Biochar yield (%db) | $p < 0.001$ | $p < 0.001$ | $p = 0.023$ |
| Energy yield (%db) | $p < 0.001$ | $p < 0.001$ | $p = 0.509$ |
| MC (%wb) | $p = 0.022$ | $p = 0.529$ | $p = 0.642$ |
| VM (%db) | $p < 0.001$ | $p < 0.001$ | $p = 0.020$ |
| FC (%db) | $p < 0.001$ | $p = 0.087$ | $p = 0.101$ |
| Ash (%db) | $p < 0.001$ | $p < 0.001$ | $p < 0.001$ |
| Fuel ratio (-) | $p < 0.001$ | $p = 0.102$ | $p = 0.194$ |
| C (%daf) | $p < 0.001$ | $p = 0.003$ | $p < 0.001$ |
| H (%daf) | $p < 0.001$ | $p < 0.001$ | $p = 0.461$ |
| N (%daf) | $p < 0.001$ | $p < 0.001$ | $p = 0.086$ |
| O (%daf) | $p < 0.001$ | $p = 0.001$ | $p = 0.001$ |
| Decarbonization (%) | $p < 0.001$ | $p < 0.001$ | $p = 0.003$ |
| Dehydrogenation (%) | $p < 0.001$ | $p < 0.001$ | $p < 0.001$ |
| Deoxygenation (%) | $p < 0.001$ | $p < 0.001$ | $p < 0.001$ |
| HHVdaf (MJ/kg) | $p < 0.001$ | $p = 0.129$ | $p < 0.001$ |
| HHVdb (MJ/kg) | $p < 0.001$ | $p < 0.001$ | $p < 0.001$ |

MC: moisture content; VM: volatile matter; FC: fixed carbon; wb: wet basis; db: dry basis; daf: dry and ash-free basis

decomposition of hemicellulose and cellulose mainly occurs at 220–315°C and 315–400°C, respectively, while that of lignin occurs in a wide temperature range of 160–900°C [26]. Furthermore, the torrefaction at 230°C just releases some moisture and light volatiles from the biomass, while that at 260°C decomposes some hemicellulose but barely affects cellulose and lignin, whereas that at 290°C destroys substantial amounts of hemicellulose and cellulose [27]. Taking the biomass compositions into consideration, the torrefactions at 210 and 250°C at 0.1 MPa mainly decomposed hemicellulose, while that of 290°C at 0.1 MPa decomposed cellulose as well as hemicellulose. In contrast, the torrefactions of 210 and 250°C under elevated pressure gave a lower yield compared to 0.1 MPa, implying that the high-pressure environment assists substantially with the decompositions of hemicellulose and cellulose even at 210 and 250°C. The same trend is also observed in rice straw and sawdust under gas-pressurized torrefaction [28]. In this study, torrefaction at 200–300°C under a pressure of 2.5 MPa decreased the biochar yields compared to the traditional torrefaction under atmospheric pressure and increased the gaseous yields of CO and $CO_2$ during the process. Since the release of CO and $CO_2$ is associated with thermal cracking of carbonyl and carboxyl groups in cellulose and hemicellulose [26], we can conclude that an elevated-pressure environment promotes the decomposition of hemicellulose and cellulose, resulting in a decrease in biochar yield.

## Proximate analysis

Proximate analyses of raw manure and biochar under different conditions are displayed in Table 2. The moisture content of the resulting chars was less than 1.0%, indicating that drying is adequate, and the handling performance in storage and transportation is markedly improved. With regard to the VM, FC, and ash content, we observed a decrease in VM and an increase in FC and ash content of the resulting chars compared to those in the raw manure throughout the process (Table 2). The statistical analyses revealed that an interaction between the temperature and pressure occurred for VM ($p = 0.020$) and ash content ($p < 0.001$), though not for FC ($p = 0.101$) (Table 1). The VM content tended to decrease, while the ash content tended to increase with increasing temperature and pressure. The difference in the VM content in the range of 0.1–1.0 MPa was 13.1%db for 210°C ($p = 0.003$), 9.2%db for 250°C

**Table 2. Proximate analyses of raw manure and biochar.** The biochar was prepared with a preheating temperature of 160°C.

| Treatment | MC (%wb) | VM (%db) | FC (%db) | Ash (%db) |
|---|---|---|---|---|
| Raw manure | 62.1 ± 0.9 | 63.7 ± 0.1 | 17.7 ± 1.1 | 18.6 ± 1.2 |
| 210°C-0.1 MPa | 0.5 ± 0.1 | 59.3 ± 0.5a | 20.4 ± 0.7 | 20.4 ± 0.2h |
| 210°C-0.4 MPa | 0.2 ± 0.0 | 47.3 ± 0.1bc | 27.8 ± 0.4 | 24.9 ± 0.5g |
| 210°C-0.7 MPa | 0.4 ± 0.6 | 46.2 ± 0.7bc | 26.5 ± 0.8 | 27.3 ± 0.1f |
| 210°C-1.0 MPa | 0.6 ± 0.2 | 46.8 ± 5.1bc | 23.1 ± 4.8 | 30.1 ± 0.3cde |
| 250°C-0.1 MPa | 0.1 ± 0.1 | 49.9 ± 0.8b | 26.3 ± 1.0 | 23.8 ± 0.2g |
| 250°C-0.4 MPa | 0.0 ± 0.0 | 41.2 ± 3.2bcde | 30.4 ± 3.3 | 28.5 ± 0.1ef |
| 250°C-0.7 MPa | 0.5 ± 0.7 | 41.7 ± 1.5bcd | 28.4 ± 0.4 | 29.8 ± 1.1de |
| 250°C-1.0 MPa | 0.0 ± 0.0 | 40.7 ± 2.4cde | 27.7 ± 1.7 | 31.6 ± 0.7c |
| 290°C-0.1 MPa | 0.0 ± 0.0 | 35.2 ± 0.8de | 34.6 ± 0.9 | 30.3 ± 0.1cd |
| 290°C-0.4 MPa | 0.0 ± 0.0 | 37.0 ± 2.0de | 31.6 ± 1.9 | 31.3 ± 0.1cd |
| 290°C-0.7 MPa | 0.0 ± 0.0 | 33.1 ± 2.3de | 33.3 ± 2.3 | 33.6 ± 0.0b |
| 290°C-1.0 MPa | 0.0 ± 0.0 | 32.4 ± 2.2e | 31.4 ± 2.1 | 36.2 ± 0.1a |

The numerical values represent mean ± standard deviation (n = 2). Different letters indicate significant differences among the means (Tukey's test, $p < 0.05$).

MC: moisture content; VM: volatile matter; FC: fixed carbon; db: dry basis

($p$ = 0.042), and 4.6%db for 290°C ($p$ = 0.642) (Table 2), indicating that elevated pressure promotes devolatilization in light or mild torrefaction (210 or 250°C). Meanwhile, its influence becomes less at higher temperatures as the torrefaction temperature becomes more dominant at severe torrefaction (290°C). In addition, the fact that there was no significant difference in the VM content in the range of 0.4–1.0 MPa suggests that even a slightly pressurized environment such as 0.4 MPa can promote devolatilization in light and mild torrefactions.

The fuel ratios of the resulting chars are displayed in Fig 4. Fuel ratio is expressed as the ratio of FC to VM and is an indicator that predicts the combustion behavior of coal and biofuels [29–31]. When introducing biomass to an existing coal power plant, it is desirable to make the fuel ratio of biomass comparable that of coal as the plant is customized for coal combustion. As can be seen in Fig 4, the fuel ratio of raw biomass was much lower than that of coal but increased throughout the process. Neither the pressure ($p$ = 0.102) nor interaction ($p$ = 0.194) affected the ratio, while it grew as temperature increased ($p$ < 0.001) (Table 1), indicating that the torrefaction temperature is a more influential parameter than ambient pressure on the fuel ratio of the resulting char. Regardless of the pressure, the fuel ratios of biochar at 290°C were around 1.0, which is almost equal to those of sub-bituminous coal [32]. Therefore, a torrefaction of 290°C can upgrade dairy manure to a solid biofuel with combustion characteristics comparable to sub-bituminous coal and therefore might be able to increase the blending ratio of biofuel at a coal power plant.

## Elemental analysis

The elemental composition of the resulting products is listed in Table 3. Overall, we observed an increase in C and N and decrease in O and H compared to those elements in raw manure. For C content, it increased from 50.8 ± 0.5%daf for raw manure to 53.1 ± 0.4–61.2 ± 0.2%daf at 210°C, 60.7 ± 0.5–64.7 ± 0.2%daf at 250°C, and 67.4 ± 1.3–71.0 ± 2.1%daf at 290°C. In contrast, the O content decreased from 40.3 ± 0.5%daf for raw manure to 29.5 ± 0.1–38.1 ± 0.4% daf at 210°C, 26.4 ± 0.2–30.4 ± 0.5%daf at 250°C, and 19.8 ± 2.2–23.3 ± 1.3%daf at 290°C. From a chemical viewpoint, the purpose of torrefaction is to remove oxygen from the biomass, increasing the relative carbon content of the resulting solid. The elemental analysis proved that

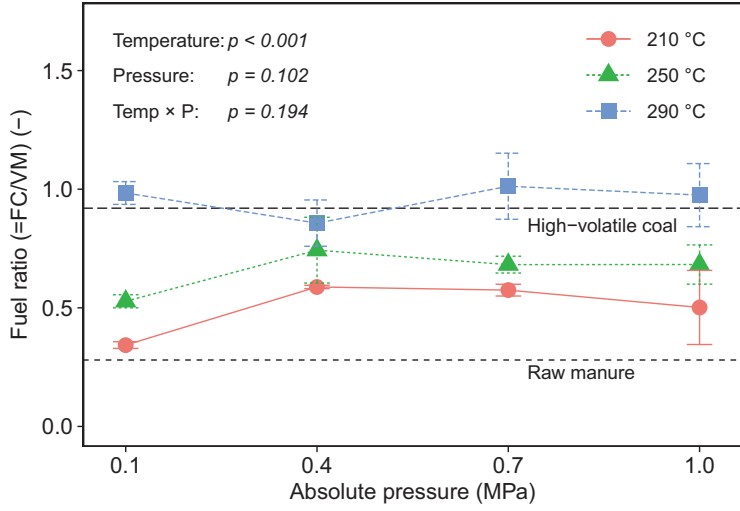

**Fig 4. Fuel ratio of biochar at torrefaction temperatures and ambient pressures.**

**Table 3. Elemental analyses of raw manure and biochar.** The biochar was prepared with a preheating temperature of 160˚C.

| Treatment | C (%daf) | H (%daf) | N (%daf) | O* (%daf) |
|---|---|---|---|---|
| Raw manure | 50.8 ± 0.5 | 6.2 ± 0.1 | 2.7 ± 0.0 | 40.3 ± 0.5 |
| 210˚C-0.1 MPa | 53.1 ± 0.4g | 5.9 ± 0.0 | 2.9 ± 0.0 | 38.1 ± 0.4a |
| 210˚C-0.4 MPa | 58.8 ± 0.6f | 4.8 ± 0.1 | 3.8 ± 0.1 | 32.6 ± 0.8b |
| 210˚C-0.7 MPa | 59.5 ± 0.7ef | 4.8 ± 0.0 | 4.2 ± 0.0 | 31.5 ± 0.7b |
| 210˚C-1.0 MPa | 61.2 ± 0.2def | 4.7 ± 0.1 | 4.7 ± 0.0 | 29.5 ± 0.1bc |
| 250˚C-0.1 MPa | 60.7 ± 0.5def | 5.5 ± 0.0 | 3.4 ± 0.0 | 30.4 ± 0.5bc |
| 250˚C-0.4 MPa | 64.7 ± 0.2bcd | 4.5 ± 0.0 | 4.4 ± 0.0 | 26.4 ± 0.2cd |
| 250˚C-0.7 MPa | 61.7 ± 1.1def | 4.5 ± 0.1 | 4.7 ± 0.2 | 29.1 ± 1.3bc |
| 250˚C-1.0 MPa | 63.7 ± 2.6cde | 4.6 ± 0.1 | 5.1 ± 0.1 | 26.6 ± 2.7cd |
| 290˚C-0.1 MPa | 71.0 ± 2.1a | 5.0 ± 0.0 | 4.3 ± 0.1 | 19.8 ± 2.2e |
| 290˚C-0.4 MPa | 68.3 ± 0.7ab | 4.1 ± 0.3 | 5.0 ± 0.0 | 22.6 ± 0.9de |
| 290˚C-0.7 MPa | 67.4 ± 1.3abc | 4.0 ± 0.1 | 5.4 ± 0.1 | 23.3 ± 1.3de |
| 290˚C-1.0 MPa | 69.5 ± 0.4a | 4.0 ± 0.2 | 6.0 ± 0.0 | 20.5 ± 0.6e |

The numerical values represent mean ± standard deviation (n = 2). Different letters indicate significant differences among the means (Tukey's test, $p < 0.05$).

daf: dry and ash-free basis

* $O = 100 - C - H - N$

the self-heating torrefaction system achieved the removal of oxygen and succeeded in producing a carbon-rich material, although the system is an oxidative process. It should be noted that the torrefaction at 290˚C and 0.1 MPa produced a biochar with the highest carbon and lowest oxygen content (Table 3). This is probably because carbon consumption is suppressed at that condition. As shown in Fig 2, drying was driven by an external heat source at 0.1 MPa, and during that time, almost no carbon was consumed due to oxidation, which may have led to the high carbon content of the biochar. Although the same phenomenon occurred at 210 and 250˚C under a pressure of 0.1 MPa, oxygen removal progresses as the torrefaction temperature increases, and as a result, the carbon content of the biochar was highest at 290˚C and 0.1 MPa.

To understand the removal of C, H, and O in biomass, three major indices, decarbonization (DC), dehydrogenation (DH), and deoxygenation (DO), were calculated according to the ash tracer method [33] and are plotted in Fig 5. As expected, the rates of DC, DH, and DO increased as the torrefaction temperature and ambient pressure increased. The extent of element removal was ranked as DO > DH > DC, indicating that more oxygen and hydrogen are removed from biomass than carbon. At the most severe condition (290˚C and 1.0 MPa), DC and DO were 45.1 ± 0.5% and 79.6 ± 0.5%, respectively, meaning that the self-heating torrefaction system can recover about 55% of the carbon from raw manure while removing about 80% of the oxygen. Similar to the trend of biochar yield, elevated pressure greatly promoted DO at a temperature of 210˚C, whereas its impact became less at higher temperatures, especially for torrefaction of 290˚C (Figs 3 and 5C). In other words, biomass decomposition involved with oxygen removal can be carried out even at 210˚C if elevated pressure is used.

A van Krevelen diagram of biochar is illustrated in Fig 6. The plot of raw manure located in the biomass region moved to the regions of peat, lignite, and coal through the self-heating torrefaction. It is known that when the severity of torrefaction is intensified, the atomic O/C and H/C ratios of biomass decrease; the plots move to the bottom-left of the diagram. The current study also confirmed that the atomic ratios of biochar became closer to those of coal at higher temperatures. With respect to pressure, the torrefaction at 210˚C and 0.1 MPa did not change the atomic ratios of manure significantly, whereas the process at 210˚C and ≥ 0.4 MPa altered

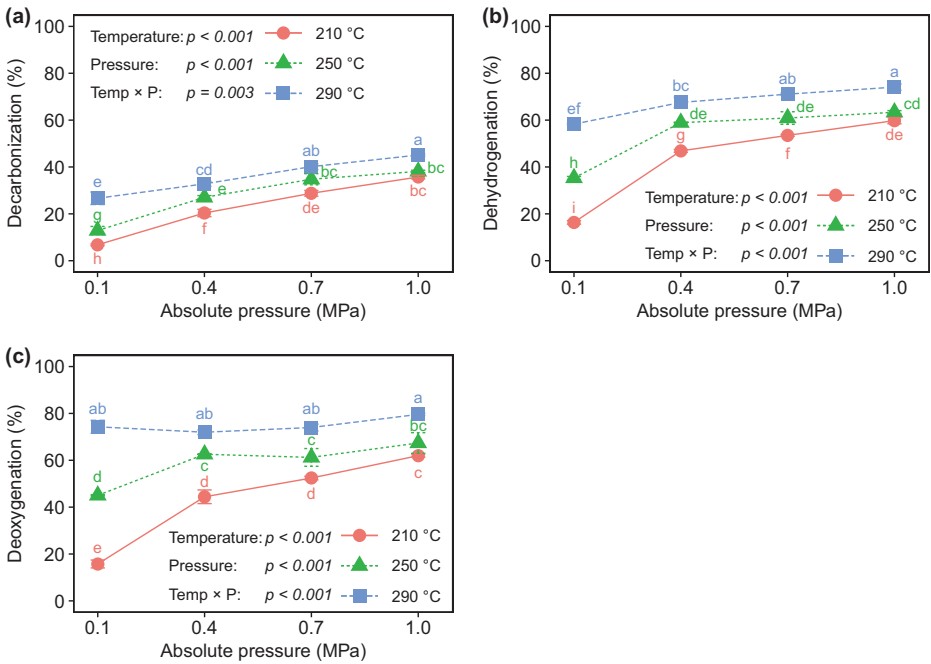

**Fig 5.** Profiles of (a) decarbonization, (b) dehydrogenation, and (c) deoxygenation of dairy manure.

manure to a peat- or lignite-like material (Fig 6). Furthermore, it is noteworthy that the biochar at 210°C and $\geq 0.4$ MPa was close to that at 250°C and 0.1 MPa. These results clearly show that the use of the elevated-pressure environment is one option to progress the coalification of the feedstock at a low torrefaction temperature. At a high torrefaction temperature, however, there was no significant difference in the atomic ratios of biochar since the impact of pressure is reduced at increasing temperature.

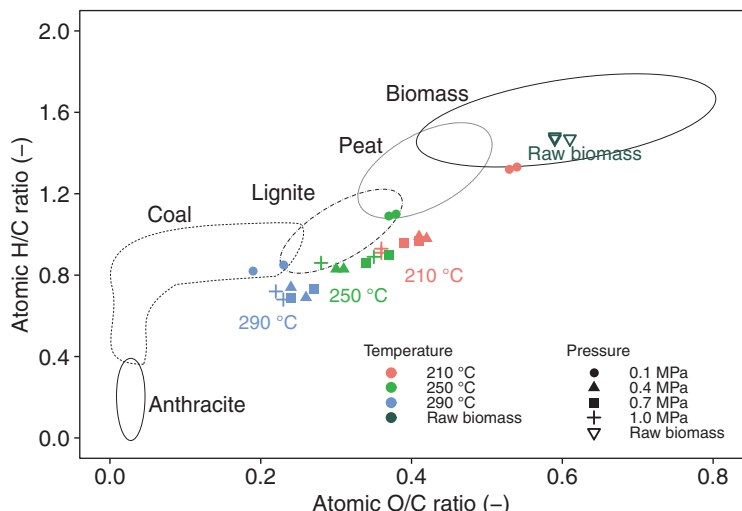

**Fig 6. A van Krevelen diagram of raw manure and biochar.** Symbols filled in red, green, and blue represent the torrefaction temperature at 210°C, 250°C, and 290°C, respectively.

## Energy analysis

The HHVs of the resulting products on a dry ash-free basis are depicted in Fig 7A. The HHVs (daf) increased with increasing temperature because of the increase in carbon content of the biochar (Fig 7A and Table 3). The elevated pressure environment increased the HHVs(daf) compared to atmospheric pressure at 210˚C, but its influence became less at 250 and 290˚C. A previous study reported that the torrefaction of wood samples at 280˚C increased the HHVs from 19.9 to 21.1 MJ/kg for aspen, 21.3 to 22.7 MJ/kg for pine, and 20.8 to 22.5 MJ/kg for beech as the pressure increased from 0.1 to 2.1 MPa [34]. The present and previous studies illustrate that elevated pressure can intensify the energy density of the organic portion of the resulting chars, but temperature is the more influential parameter. In the present study, we observed a maximum HHV(daf) of 28.6 MJ/kg at 290˚C and 0.1 MPa owing to it having the highest carbon content (Fig 7A and Table 3). This is because the substantial length of the process was driven by external heat (Fig 2), thereby resulting in the prevention of carbon consumption. On the other hand, the HHVs of the resulting products on a dry basis exhibited the opposite trend; the HHV(db) of biochar decreased as pressure increased (Fig 7B). This is due to the enlargement of the relative ash content in the resulting products (Table 3), because the absolute amount of ash hardly changes through the process while the organic portion is reduced. This phenomenon is not limited to self-heating torrefaction. The conventional torrefaction of sewage sludge also shows a downward trend due to the increase in ash content when higher temperatures and longer residence times are used [35]. Accordingly, excessive elevated pressure should be avoided when high-ash feedstock is used.

The preheating temperature for the process also affects the calorific values of the solid products. Considering the calorific values of raw manure (16.6 MJ/kg for HHV[db] and 20.2 MJ/kg for HHV[daf]), the use of a preheating temperature of 160˚C resulted in increased energy density of the products (Fig 7A and 7B). Our previous study using a preheating temperature of 100˚C observed significant drops in HHV(db) of 14.6–15.8 MJ/kg compared to raw manure [20]. Since the system oxidizes the feedstock to raise its temperature, the use of a lower preheating temperature tends to decompose biomass more, while that of a higher preheating temperature helps save the consumption of carbon during the early stage of the process. The comparison revealed that the preheating temperature affects the chemical properties of resulting chars and is therefore a parameter that should be considered for further system optimization.

We calculated the energy yield, which is determined by biochar yield and HHVs of feedstock and biochar as reported in the literature [36]. Energy yield means the fraction of the original energy in the raw biomass retained after torrefaction and is used to identify torrefaction

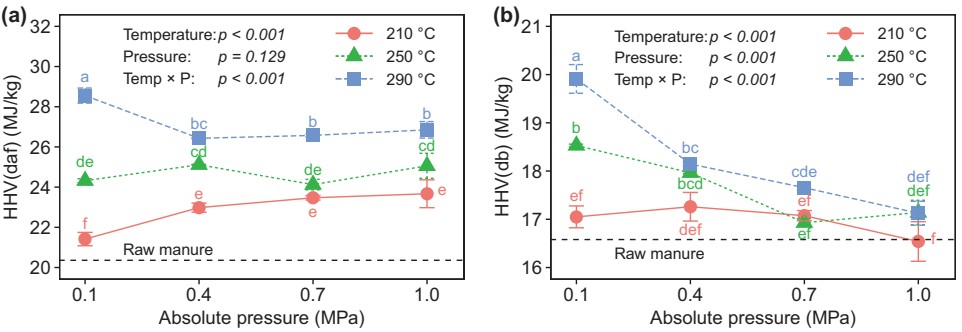

**Fig 7.** The higher heating values of biochar in (a) a dry and ash-free basis and (b) a dry basis.

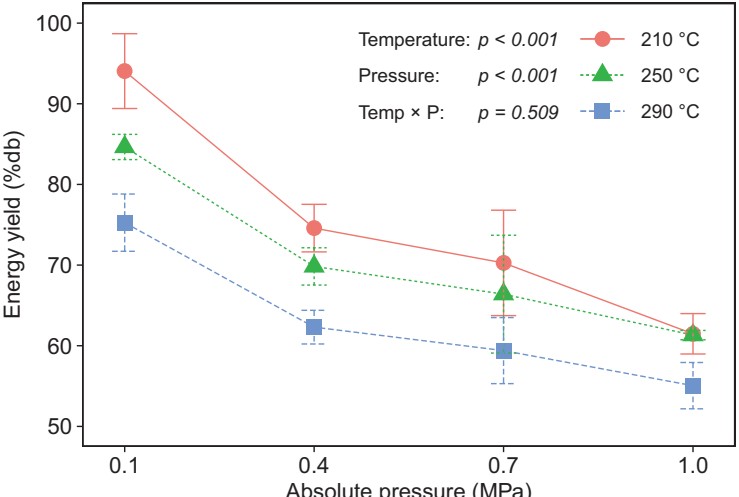

**Fig 8. Energy yields under different torrefaction temperatures and ambient pressures.**

performance [37]. The energy yields decreased with increasing torrefaction severity and were similar to the trends of biochar yield (Figs 3 and 8). This suggests that biochar yield is the dominant factor in determining the energy yield. In the current study, we observed that the lowest energy yield was $55.1 \pm 2.9\%$ at 290°C and 1.0 MPa, indicating that at least 55.1% of the original energy in the raw manure was preserved in the biochar. According to a comparative study between hydrothermal carbonization and pyrolysis of poultry litter, the energy yields varied over 46.3–54.0% for hydrochar at 175–250°C and 36.6–77.5% for biochar at 250–500°C [38]. The ranges of energy yield obtained in the present study were compatible with conventional conversion processes, implying that the self-heating torrefaction system has potential for waste-to-energy conversion.

## Conclusions

We investigated the initiating temperatures of self-heating of manure and the chemical characteristics of biochar under different pressures, intending to understand the role of ambient pressure in a self-heating torrefaction system. Elevated pressure lowered the initiating temperature for self-heating of manure because it helped supply more oxygen and retain more moisture to promote chemical oxidation. Elevated pressure also promoted the decomposition of manure, as well as devolatilization, deoxygenation, and coalification, and intensified the energy density of the organic portion of resulting chars compared to atmospheric pressure at lower torrefaction temperatures, but its influence became less at higher torrefaction temperatures. These results therefore indicate that ambient pressure considerably affects the self-heating torrefaction system and is an important parameter for further system optimization.

## Acknowledgments

We express our gratitude to Ms. Ai. Tokumitsu and Ms. Satomi Sawasato, Instrumental Analysis Division, Global Facility Center, Creative Research Institution, Hokkaido University for performing elemental analyses using a CE-440 elemental analyzer.

## Author Contributions

**Conceptualization:** Kazunori Iwabuchi, Katsumori Taniguro.

**Formal analysis:** Takanori Itoh.

**Funding acquisition:** Takanori Itoh, Kazunori Iwabuchi.

**Investigation:** Takanori Itoh, Naohiro Maemoku, Siyao Chen.

**Methodology:** Takanori Itoh.

**Project administration:** Kazunori Iwabuchi.

**Supervision:** Kazunori Iwabuchi.

**Writing – original draft:** Takanori Itoh.

**Writing – review & editing:** Kazunori Iwabuchi.

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
