## [Decision Letter · Decision Letter 0]

19 Feb 2020

PONE-D-20-01902

Role of ambient pressure in self-heating torrefaction of dairy cattle manure

PLOS ONE

Dear Prof. Iwabuchi,

Thank you for submitting your manuscript to PLOS ONE. After careful consideration, we feel that it has merit but does not fully meet PLOS ONE’s publication criteria as it currently stands. Therefore, we invite you to submit a revised version of the manuscript that addresses the points raised during the review process.

We would appreciate receiving your revised manuscript by Apr 04 2020 11:59PM. To enhance the reproducibility of your results, we recommend that if applicable you deposit your laboratory protocols in protocols.io, where a protocol can be assigned its own identifier (DOI) such that it can be cited independently in the future. For instructions see: http://journals.plos.org/plosone/s/submission-guidelines#loc-laboratory-protocols

We look forward to receiving your revised manuscript.

Kind regards,

Paulo H. Pagliari

Academic Editor

PLOS ONE

Journal Requirements:

1. Thank you for including your competing interests statement; "This study received financial support from Tanigurogumi Corporation, Japan. T.I. has been an employee of Tanigurogumi Corporation, Japan, since April 2019. There is no further employment, consultancy, patents, products in development, and marketed products to declare. This does not alter our adherence to PLOS ONE policies on sharing data and materials."

We note that one or more of the authors is affiliated with the funding organization, indicating the funder may have had some role in the design, data collection, analysis or preparation of your manuscript for publication; in other words, the funder played an indirect role through the participation of the co-authors.

If the funding organization did not play a role in the study design, data collection and analysis, decision to publish, or preparation of the manuscript and only provided financial support in the form of authors' salaries and/or research materials, please review your statements relating to the author contributions, and ensure you have specifically and accurately indicated the role(s) that these authors had in your study in the Author Contributions section of the online submission form. Please make any necessary amendments directly within this section of the online submission form.  Please also update your Funding Statement to include the following statement: “The funder provided support in the form of salaries for authors [insert relevant initials], but did not have any additional role in the study design, data collection and analysis, decision to publish, or preparation of the manuscript. The specific roles of these authors are articulated in the ‘author contributions’ section.”

If the funding organization did have an additional role, please state and explain that role within your Funding Statement.

Please also provide an updated Competing Interests Statement declaring this commercial affiliation along with any other relevant declarations relating to employment, consultancy, patents, products in development, or marketed products, etc.  

Additional Editor Comments (if provided):

The manuscript is missing critical important information which must be added before the manuscript can be considered for publication. The use of replicate for the data presented as well as how the data was statistically analyzed must be present. Otherwise the manuscript is written in very good English and it is very easy to follow and understand. The only drawback is the methodology section missing the above mentioned info.

Reviewers' comments:

Reviewer's Responses to Questions

**Comments to the Author**

1. Is the manuscript technically sound, and do the data support the conclusions?

Reviewer #1: Partly

2. Has the statistical analysis been performed appropriately and rigorously? 

Reviewer #1: No

3. Have the authors made all data underlying the findings in their manuscript fully available?

Reviewer #1: No

4. Is the manuscript presented in an intelligible fashion and written in standard English?

Reviewer #1: Yes

5. Review Comments to the Author

Reviewer #1: Title: Role of ambient pressure in self-heating torrefaction of dairy cattle manure

Author: Takanori Itoh, Kazunori Iwabuchi, Naohiro Maemoku, Katsumori Taniguro

Manuscript ID: PONE-D-20-01902

Overall Comments:

This research investigated the effect of ambient pressure in self-heating torrefaction of livestock manure. Truly, this paper is a well written article, easy to follow, and showed sequential flow of information, and the authors have demonstrated good knowledge of the overall problem. This is a nice piece of work and can be published after incorporating the following comments.

Principal Comments:

Material and Methods section is short and lacking sufficient information. Authors are strongly advised to include name of all standard methods (ASTM- American Society for Testing and Materials) followed to perform ultimate and proximate analyses along with a very brief description of the measurement methods and instruments used. Example are as follows:

Standard Methods Instruments/ Model Measurement methods

Proximate

Volatile Matter ASTM D7582 Leco TGA 701 Thermogravimetric Analyzer Mass loss (% dry basis) of the specimen when heated at 950ºC in presence of oxidizing gas for 2 h.

Heating Value

(Calorific Value) ASTM D5865 Parr 6200 Calorimeter, Parr Instruments (Moline, IL, USA) Oxygen bomb calorimeter

Statistical analysis: A small section of statistical methods and analysis should be included. Nothing is mentioned about replications whether samples were replicated or duplication or subsamples. To publish this article, sample should be duplicated. This is major drawback of this article.

6. PLOS authors have the option to publish the peer review history of their article (what does this mean?). If published, this will include your full peer review and any attached files.

Reviewer #1: No

---

## [Author Response · Author response to Decision Letter 0]

30 Mar 2020

Comment from Academic Editor:

The manuscript is missing critical important information which must be added before the manuscript can be considered for publication. The use of replicate for the data presented as well as how the data was statistically analyzed must be present. Otherwise the manuscript is written in very good English and it is very easy to follow and understand. The only drawback is the methodology section missing the above mentioned info.

Response:

We reperformed experiments in the revised version of the manuscript to ensure reproducibility. The average values ± standard deviations of two replicates are reported. The obtained data were subjected to a two-way ANOVA followed by Tukey’s test. All data on the properties of biochar have been updated, but these changes did not affect the interpretation of the results and conclusions. We added this information as well as an improved explanation of the experimental design in the Materials and methods section.

Comment from Reviewer #1:

Principal Comments: Material and Methods section is short and lacking sufficient information. Authors are strongly advised to include name of all standard methods (ASTM- American Society for Testing and Materials) followed to perform ultimate and proximate analyses along with a very brief description of the measurement methods and instruments used. Example are as follows: Standard Methods Instruments/ Model Measurement methods

Proximate

Volatile Matter ASTM D7582 Leco TGA 701 Thermogravimetric Analyzer Mass loss (% dry basis) of the specimen when heated at 950ºC in presence of oxidizing gas for 2 h.

Heating Value

(Calorific Value) ASTM D5865 Parr 6200 Calorimeter, Parr Instruments (Moline, IL, USA) Oxygen bomb calorimeter

Response:

We appreciate your advice. Although we added the instrument for VM determination along with a description of the measurement method in Sample Analyses, we cannot show the names of all standard methods for proximate, ultimate, and calorific analyses because the analyses were not strictly based on standard methods. However, as all the analyses were performed by methods similar to the standard method (ASTM E871-82, ASTM E1755-01, and ASTM D2015 for moisture, ash, and calorific value), we believe that the data obtained are reliable.

Statistical analysis: A small section of statistical methods and analysis should be included. Nothing is mentioned about replications whether samples were replicated or duplication or subsamples. To publish this article, sample should be duplicated. This is major drawback of this article.

Response:

We added explanations of the statistical analyses as well as the experimental design in the Materials and methods section.

---

## [Decision Letter · Decision Letter 1]

28 Apr 2020

Role of ambient pressure in self-heating torrefaction of dairy cattle manure

PONE-D-20-01902R1

Dear Dr. Iwabuchi,

We are pleased to inform you that your manuscript has been judged scientifically suitable for publication and will be formally accepted for publication once it complies with all outstanding technical requirements.

With kind regards,

Paulo H. Pagliari

Academic Editor

PLOS ONE

Additional Editor Comments (optional):

Reviewers' comments:

Reviewer's Responses to Questions

**Comments to the Author**

1. If the authors have adequately addressed your comments raised in a previous round of review and you feel that this manuscript is now acceptable for publication, you may indicate that here to bypass the “Comments to the Author” section, enter your conflict of interest statement in the “Confidential to Editor” section, and submit your "Accept" recommendation.

Reviewer #2: All comments have been addressed

2. Is the manuscript technically sound, and do the data support the conclusions?

Reviewer #2: Yes

3. Has the statistical analysis been performed appropriately and rigorously? 

Reviewer #2: Yes

4. Have the authors made all data underlying the findings in their manuscript fully available?

Reviewer #2: Yes

5. Is the manuscript presented in an intelligible fashion and written in standard English?

Reviewer #2: Yes

6. Review Comments to the Author

Reviewer #2: The manuscript has scientific interest because the used torrefaction system, which uses the heat generated from the chemical oxidation of manure, is a new approach.

7. PLOS authors have the option to publish the peer review history of their article (what does this mean?). If published, this will include your full peer review and any attached files.

Reviewer #2: Yes: Dr.Jale YANIK

---

## [Editor Report · Acceptance letter]

20 May 2020

PONE-D-20-01902R1 

Role of ambient pressure in self-heating torrefaction of dairy cattle manure 

Dear Dr. Iwabuchi:

I am pleased to inform you that your manuscript has been deemed suitable for publication in PLOS ONE. Congratulations! Your manuscript is now with our production department. 

With kind regards,

on behalf of

Dr. Paulo H. Pagliari 

Academic Editor

PLOS ONE